# Forest Land Resource Information Acquisition with Sentinel-2 Image Utilizing Support Vector Machine, K-Nearest Neighbor, Random Forest, Decision Trees and Multi-Layer Perceptron

**Chen Zhang [1], Yang Liu [1] and Niu Tie [2,\*]**

1   College of Forestry, Inner Mongolia Agriculture University, Hohhot 010018, China
2   Forestry and Grassland Bureau of Inner Mongolia Autonomous Region, Hohhot 010020, China
*   Correspondence: fengzili@emails.imau.edu.cn

**Abstract:** Forestry work involves scientific management and the effective utilization of forest land resources, and finding economical, efficient and accurate acquisition methods for forest land resource information. In previous land-use classification research, machine learning algorithms have achieved good results, and Sentinel public data have been used in various remote sensing applications. However, there is a paucity of research using these data to evaluate the performance of machine learning algorithms in the extracting of complex forest land resource information. Using the Sentinel-2 satellite multispectral image data, the spectral reflectance, vegetation index characteristics and image texture characteristics of different forest land resources in the study area were calculated and compared. Then, based on three groups of features, the performances of the Support Vector Machine (SVM), K-Nearest Neighbor (KNN), Random Forest (RF), decision trees (DT) and Multi-layer Perceptron (MLP) were examined and compared to identify and classify forest land resource types. The research indicates the following: (1) The SVM algorithm achieved the highest OA (95.8%). The average accuracy of the SVM algorithm was much higher than other algorithms (SVM 88.3%, KNN 87.5, RF 85.3%, MLP 85.00% and DT 77.5%). (2) The classification accuracies of each algorithm for coniferous forests were relatively high, and the recognition accuracy was above 95%, whereas the classification accuracies of the other categories varied greatly. (3) Adding texture features can improve the accuracy of the five algorithms. This study reports new references for the qualitative methods of forest land resource distribution. It has also produced more efficient and accurate acquisitions of forest land resource information, scientific management and effective use of forest land resources.

**Keywords:** forest land resource information; sentinel-2; classification algorithms; machine learning

## 1. Introduction

Forest ecosystems have an important impact on climate change [1]. Therefore, it is of great significance to study the qualitative methods of forest land resource distribution to understand forest dynamics and evaluate the climate. In order to scientifically manage and effectively utilize forest land resources, forestry work focuses on finding an economical, efficient and accurate method for obtaining information on forest land resources. The rapid development of space remote sensing technology has provided an effective means for obtaining information on national forest land resources. The regular and real-time monitoring of forest resources, and the accurate, fast, high-quality and efficient acquisition of information from different sources and different forms, all provide a useful tool for inventorying forest resources, forecasting forest fires and utilizing and protecting forest resources [2]. At present, scholars have carried out much research on the extraction of forest land resources by using different remote sensing data to improve the classification accuracy of forest land resources [3].

Sentinel-2 carries a multispectral imager (MSI) for land monitoring, which can provide images of vegetation, soil and water coverage, inland waterways and coastal areas, with a

high spatial resolution and good spectral quality [4,5]. With the free disclosure of data, a variety of new remote sensing research possibilities have emerged. In previous studies on forest vegetation, the three red-edge vegetation and Shortwave-infrared (SWIR) bands in Sentinel-2 are more sensitive to chlorophyll content and enable the distinction of different vegetation types and Land-Use Land Cover (LULC) classification accuracy [6]. Nelson [7] classified the tree populations in central Sweden through multi-time-series Sentinel-2 data and Random Forest (RF) classifiers (including mixed coniferous forests, coniferous and deciduous forests, deciduous and broad-leaved forests, etc.). Hawrylo et al. [8] used Sentinel-2 data to test the RF and Support Vector Machine (SVM) algorithms by investigating the defoliation of Scots pines in Poland, and found that Sentinel-2 data are suitable for this goal. In addition, studies have found that spectral features and texture features can effectively improve the accuracy of the vegetation species classification and can be used to distinguish different forest species [9,10]. Several studies found that Sentinel-2 data have a high potential for use in different classification tasks and applications, such as tree species classification [11–13], information extraction of burned areas, forest-type classification, etc. [14–17].

The improvement of land-use classification results not only depends on the suitability of remote sensing images, but also the correct selection of classification methods [18]. In recent years, the continuous development of remote sensing technology and the rise of machine classification algorithms have accelerated the intelligent process of image recognition. At present, many advanced machine integration algorithms and classifiers have been applied to remote sensing image classification, and these methods have been successful in land use mapping and monitoring [19–21]. Hatami et al. found that in remote sensing image classification, machine learning algorithms such as SVM, K-Nearest Neighbor (KNN) and RF are superior to other traditional supervised classifiers [22]. In recent years, the cellular neural network (CNN) has generally achieved better classification performance compared to other types of deep learning. However, Multi-layer Perceptron (MLP), a basic neural network, has proven to be a promising machine learning technique [23–25]. For example, compared with CNN, Xin He and Yushi Chen [26] improved the results of hyperspectral image classification by improving the MLP model, indicating that MLP-based methods are still competitive in remote sensing image classification.

Therefore, research should compare and evaluate the performance of SVM, KNN, RF, decision trees (DT) and MLP for forest land resource information acquisition methods in the south of Genhe City (Located in Inner Mongolia Autonomous Region, China) using the new satellite data and Sentinel-2 images. The objectives of this study are: (i) to evaluate the performance of the five classifiers, SVM, KNN, RF, DT and MLP, when applied to a Sentinel-2 image and (ii) to assess the effects of the spectral reflectance, vegetation index characteristics and image texture characteristics on the accuracy of the forest resource information extraction results of the five aforementioned classifiers.

## 2. Materials and Methods

### 2.1. Study Area

Genhe City is a county-level city in the north of Hulunbeir City, the Inner Mongolia Autonomous Region, located in the northern section of the Great Khingan Mountains. It has one of the highest latitudes of all cities in China, and is also the county-levelcity with the lowest average temperature in China. The resources of Genhe City are mainly forest resources, with a forest coverage rate of 91.7% and a forest area of 1.745 million hm$^2$. Pinus sylvestris var.mongolica, Larix gmelinii and Betula platyphylla are the main tree species in the area. The core area of this study is rectangular (50°59′–50°52′ N, 121°24′–121°35′ E), and it includes the first forest ecological observation station in Inner Mongolia's "Greater Khingan Mountains Forest Climate Ecological Observation Station", which is located in the south of Genhe City (Figure 1). The study area mainly includes six typical classes: broad-leaved forests, shrubland, barren land, impervious surface, grasslands and coniferous forests.

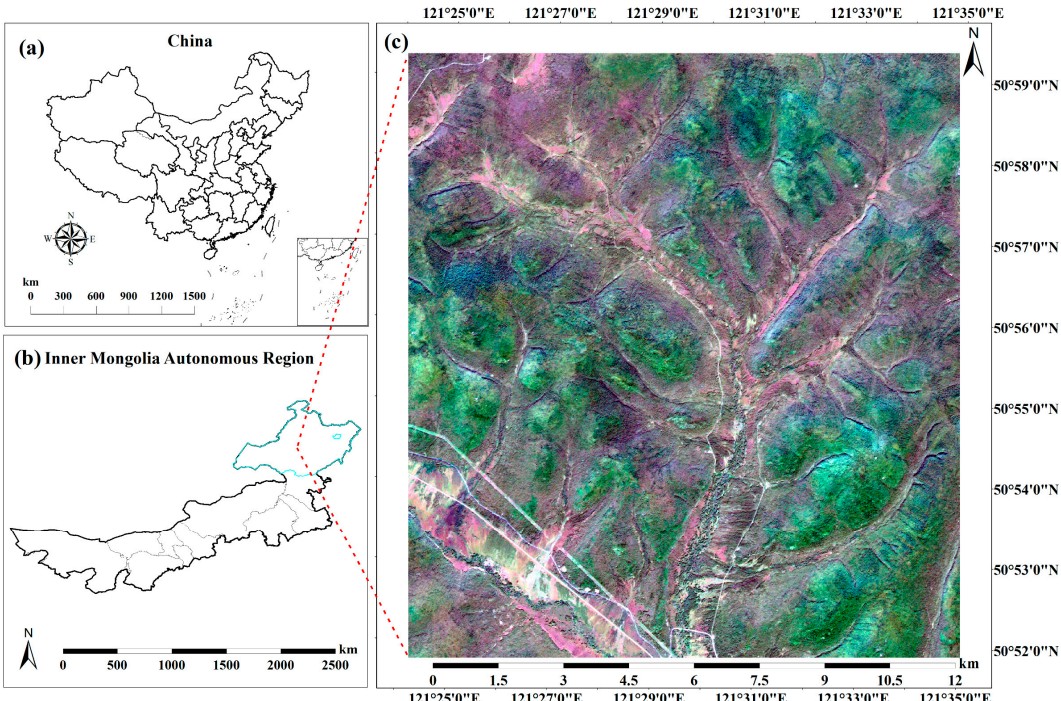

**Figure 1.** The study area: (**a**) China (**b**) Inner Mongolia Autonomous Region, with blue polygons representing Hulunbeir, (**c**) the Research Area.

### 2.2. Data Used

The multispectral image data used in this study are from the Sentinel-2 platform, downloaded from the ESA Copernicus Data Center (https://scihub.copernicus.eu, accessed on 1 September 2022). The satellite revisit period was 5 days. The width was 290 km, covering 13 spectral bands from visible light to short-wave infrared, with a spatial resolution of up to 10 m [27]. This paper selects the L2A-level data product of Sentinel-2 located in Genhe City (1 September 2022) as the research area. The geographic coordinate system used was UTM/WGS84. Due to image distortion problems caused by remote sensors, sun height, atmospheric scattering, etc., the reflectivity data of the lower atmosphere in the study area can be directly obtained. In this paper, the three bands unrelated to vegetation growth (Band-1, Band-9 and Band-10) were removed. For the image data of the remaining bands, the sen2cor tool provided by ESA was used to preprocess the image, by using radiometric calibration, atmospheric correction, resampling, format conversion, and band synthesis. Multispectral data with a resolution of 10 m were generated, and spatial registration and vector clipping were performed in the ENVI software. The parameter information of each band is shown in Table 1.

**Table 1.** Multispectral band parameters of sentinel-2 satellite.

| Band | Central Wavelength (nm) | Bandwidth (nm) | Spatial Resolution (m) |
|---|---|---|---|
| 2-Blue | 443.9 | 98 | 10 |
| 3-Green | 560.0 | 45 | 10 |
| 4-Red | 664.5 | 38 | 10 |
| 5-Red Edge | 703.9 | 19 | 20 |
| 6-Red Edge | 740.2 | 18 | 20 |
| 7-Red Edge | 782.5 | 28 | 20 |
| 8-NIR | 835.1 | 145 | 10 |
| 8A-Red Edge | 864.8 | 33 | 20 |
| 11-SWIR-1 | 1613.7 | 143 | 20 |
| 12-SWIR-2 | 2202.4 | 242 | 20 |

### 2.3. Feature Setting

The categorical features in this study were divided into three groups:

(1) Multispectral features of Sentinel-2 images (Mul); (2) Combined characteristics of Multispectral features and vegetation index characteristics (Mul-vegetation); and (3) Combination of multispectral features and texture features of Grey-Level Cooccurrence Matrix (Mul-GLCM). Based on the above three sets of characteristics, the performance of four machine learning algorithms, SVM, KNN, RF, DT and MLP, in the extraction of forest resources information was explored.

Based on the multispectral images of the study area, this study selected some commonly used vegetation indices for calculation, including: Normalized Difference Vegetation Index (NDVI), Green Red Vegetation Index (GRVI), Difference Vegetation Index (DVI), Ratio Vegetation Index (RVI), Normalized Difference Red-Edge Index (NDREI) and Land Surface Water Index (LSWI). Based on the Grey-Level Co-occurrence Matrix (GLCM) [19], the texture features of Sentinel-2 multispectral images were calculated, the filter window size was set to 3 × 3, and the mean, variance, synergy, contrast, dissimilarity and information entropy of image texture were calculated. Second-order moment and dissimilarity features were used for further analysis of texture features. The vegetation index and texture feature details are presented in Table 2.

**Table 2.** Vegetation Index and Texture Feature Details.

| Feature Types | Feature Names | Details | Remarks |
|---|---|---|---|
| Vegetation indices | Ratio vegetation index (RVI) | NIR/R | / |
| | Difference vegetation index (DVI) | NIR − Blue | |
| | Normalized difference vegetation index (NDVI) | (NIR1 − R)/(NIR1+ R) | |
| | Green Red Vegetation Index (GRVI) | (Green − R)/(Green + R) | |
| | Normalized Difference Red-Edge I Index (NDRE I) | (Red-edge 2 − Red-edge 1)/(Red-edge 2 + Red-edge 1) | |
| | Land Surface Water Index (LSWI) | (NIR − SWIR-1)/(NIR + SWIR-1) | |
| Texture features based on the gray-level co-occurrence matrix (GLCM) | Mean (ME) | $\sum_{i=0}^{N-1}\sum_{j=0}^{N-1} P(i,j)*i$ | $P(i,j) = V(i,j)/\sum_{i=0}^{N-1}\sum_{j=0}^{N-1} V(i,j)$ |
| | Variance (VA) | $\sum_{i=0}^{N-1}\sum_{j=0}^{N-1}(i-mean)^2 P(i,j)$ | $V(i,j)$ is the $i$th row of the $j$th column in the $N$th moving window |
| | Entropy (EN) | $-\sum_{i=0}^{N-1}\sum_{j=0}^{N-1} P(i,j)\log(P(i,j))$ | $u_x= \sum_{j=0}^{N-1} j \sum_{i=0}^{N-1} P(i,j)$ |
| | Angular second moment (SE) | $\sum_{i=0}^{N-1}\sum_{j=0}^{N-1} P(i,j)^2$ | $u_y= \sum_{i=0}^{N-1} i \sum_{j=0}^{N-1} P(i,j)$ |
| | Homogeneity (HO) | $\sum_{i=0}^{N-1}\sum_{j=0}^{N-1} \frac{P(i,j)}{1+(i-j)^2}$ | $\sigma_x= \sum_{j=0}^{N-1}(j-u_i)^2 \sum_{i=0}^{N-1} P(i,j)$ |
| | Contrast (CON) | $\sum_{\|i-j\|=0}^{N-1}\|i-j\|^2 \left\{ \sum_{i=1}^{N}\sum_{j=1}^{N} P(i,j) \right\}$ | $\sigma_y= \sum_{i=0}^{N-1}\left(i-u_j\right)^2 \sum_{j=0}^{N-1} P(i,j)$ |
| | Dissimilarity (DI) | $\sum_{\|i-j\|=0}^{N-1}\|i-j\| \left\{ \sum_{i=1}^{N}\sum_{j=1}^{N} P(i,j) \right\}$ | |
| | Correlation (COR) | $\frac{\sum_{i=0}^{N-1}\sum_{j=0}^{N-1} P(i,j)^2 - \mu_x\mu_y}{\sigma_x\sigma_y}$ | |

### 2.4. Training Sample Datasets

The training dataset was collected from raw Sentinel-2 data, and a visual interpretation of high-resolution imagery was provided by Google Earth. To collect training sample data, the Regions of Interest tool in the ENVI 5.6 Toolbox was used to create 50 polygons for the forest land resource information. Due to the different polygon sizes, the number of pixels for each land cover class was also different. Details are shown in Table 3.

**Table 3.** Training Datasets and Validation Datasets.

| Land Cover | Training Datasets (Objects) | Training Datasets (Pixel) |
|---|---|---|
| Broad-leaved forests | 50 | 691 |
| Shrubland | 50 | 478 |
| Barren land | 50 | 507 |
| Impervious surface | 50 | 504 |
| Grasslands | 50 | 529 |
| Coniferous forests | 50 | 653 |

### 2.5. Machine Learning Image Classification

In this study, five machine learning algorithms were used to perform pixel-based supervised image classification: SVM, KNN, RF, DT and MLP.

The main process includes the following four parts: (1) Preprocessing of sentry image data and the extraction of the vegetation index and texture information; (2) The Mul, Mul vegetation and Mul GLCM feature settings, and the division of the training set and test set; (3) The parameters of SVM, KNN, RF, DT and MLP are adjusted, and the classification results under the best parameters of each classifier are used to compare the performance of the classifier; (4) Model accuracy verification and result analysis. Details are shown in Figure 2.

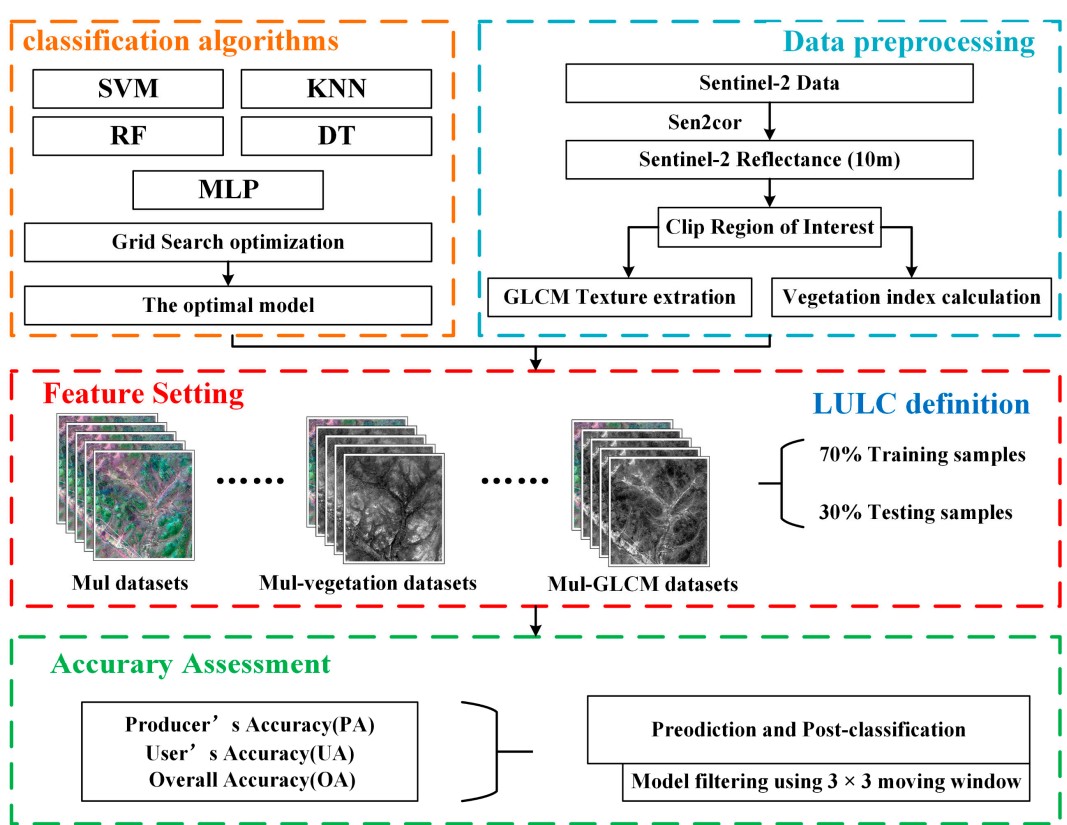

**Figure 2.** The methodological workflow.

### 2.5.1. Support Vector Machine (SVM)

The fundamental principle of SVM learning is to solve the separation hyperplane that can correctly divide the training data sets with the largest geometric interval. It transforms the nonlinear classification problem into a high-dimension linear problem, and constructs a linear discriminant function in the high-dimension feature space and introduces a kernel function to reduce the amount of calculation [28,29]. According to previous studies, despite the different types of kernels in kernel functions, including linear, polynomial, radial basis

function and sigmoid, the radial basis function (RBF) kernel is the most effective parameter in remote sensing image classifications [30–32]. To exploit the potential of RBF kernels, the penalty value (C) and gamma ($\gamma$) should be optimized. The C parameter balances the relationship between the complexity of the support vector and the misclassification rate. The larger the C, the poorer the generalization ability, which causes the overfitting phenomenon; the smaller the C, the better the generalization ability, which can also cause the overfitting phenomenon. The $\gamma$ mainly defines the influence of a single sample on the entire classification hyperplane. When $\gamma$ is relatively large, a single sample has a greater impact on the entire classification hyperplane and is more likely to be selected as a support vector, or the entire model will have more support vectors [29]. Because of the research of Li et al. [21], in this study, the parameters of RBF were set as C value = 30 and gamma = 0.0001. This procedure was applied to all datasets.

### 2.5.2. K-Nearest Neighbor (KNN)

KNN is the simplest classification algorithm. The algorithm relies on the distance between eigenvectors. The basic theory is that it finds a group of k samples closest to unknown samples in the dataset. From the k samples, the label of the unknown sample is determined by computing the mean of the response variables [33,34]. The KNN classification algorithm has two key parameters: "n_ neighbors" and "Weights" [35]. "n_ neighbors" indicates the number of nearest neighbors to be used in the learning process, that is, the k value; "Weights" is used to identify the weight of each sample's neighbor samples. The Weight function is used in the prediction of possible values: "uniform" means that all nearest neighbor samples have the same weight; "distance" indicates that the weight and distance are inversely proportional [36]. To optimize results, Scikit-Learn's GridSearchCV is used to systematically traverse multiple parameter combinations and determine the best effect parameters through ten-fold cross-validation; multiple k-values and distance metrics (uniform and distance) were included in the GridSearchCV parameters. In this study, the most accurate results for the KNN algorithm were obtained when the n_ neighbors was set to 4, while the Weights were set to "distance".

### 2.5.3. Random Forest (RF)

RF is an integrated classifier, which obtains different training sample sets by resampling samples, trains decision trees on these new training sample sets, and finally, merges the results of each learner. The final classification/prediction decision is based on majority voting, and the result is the classification label with the most "votes" [37]. RF classifiers use the Gini index as an attribute-selection criterion, which measures the heterogeneity of attributes associated with classes [38]. There are two important parameters affecting the RF classification accuracy: the number of trees (Ntree) and the number of features in each split (Mtry) [39]. According to previous research results on these two parameters, the most common suggestion is to set the Ntree parameter to 500 and set Mtry to the square root of the number of input variables. To find the optimal RF model for classification, a range of values for Ntree ('n_estimators') and Mtry ('max_features') were tested and evaluated in this study: n_estimators of 1 to 1000 and max_features count from 1 to 10. It can be seen from the test results that the RF classifier performs best when n_estimates are set to 500 and max_feature is set to 5.

### 2.5.4. Decision Trees (DT)

DT is a tree structure that describes the classification of instances [40,41]. In remote sensing image classification, decision trees effectively use the probability distribution defined in feature and class space to construct models. The algorithm has the advantages of fast classification speed, a concise model, readability, and easy implementation [42,43]. DT construction usually includes feature selection, the generation of decision trees and the pruning of decision trees [44]. Feature selection calculates the Info Gain Ratio of all variables and selects the optimal variable with a high value as the split dataset. DT

organizes information extracted from data sets into a recursive hierarchical structure made up of nodes and branches. The main process of decision tree classification starts from the root node, then tests a certain feature of the instance, and finally assigns the instance to its child nodes according to the test results; at this time, each child node corresponds to a value of the feature. The instances are tested and assigned recursively until the leaf nodes are reached, and then, the instances are classified into the classes of the leaf nodes [20,41,42]. Branches are pruned when a node's weighted error exceeds that of its parent. Finally, an optimal decision tree is generated. To optimize results, Scikit-Learn's GridSearchCV is used to systematically traverse multiple parameter combinations and determine the most effective parameters through ten-fold cross-validation. In this study, the most accurate results for the DT algorithm were obtained when the 'criterion' was set to gini, 'max_depth' was set to 7, 'min_impurity_decrease' was set to 0.1 and 'min_samples_leaf' was set to 5.

### 2.5.5. Multi-Layer Perceptron (MLP)

MLP is a forward-structured artificial neural network (ANN) that maps a set of input vectors to a set of output vectors [24]. An MLP can be viewed as a directed graph consisting of multiple layers of nodes, each fully connected to the next layer. Except for the input node, each node is a neuron with a non-linear activation function. The main process [45] includes (1) the random distribution of the weights of all edges; (2) forward propagation: using the input features of all samples in the training set as the input layer, the ANN is activated for all inputs in the training data sets, and then passes through forward propagation to obtain the output value; (3) backpropagation: using the output value and sample value to calculate the total error, and then, using backpropagation to update the weight; (4) repeat 2–3 times until the output error is lower than the established standard. In this study, the neural network models (supervised) module in Scikit-Learn was used to construct the MLP model. Through multiple experiments, the main parameters of the model were set as follows: the hidden layer was set to seven layers (64, 128, 256, 512, 256, 128, 64); the activation function of the hidden layer was set to "relu"; the solver for weight optimization was set as 'adam', which works relatively well on relatively large datasets (i.e., those with over a thousand training samples or more), in terms of both training time and validation score; alpha was set at 0.01.

### 2.6. Accuracy Assessment and Comparisons

Because the study area has implemented the policy of closing mountains for afforestation, the environment in the forest is relatively complex. The verification sample datasets are mainly field survey data, although it also refers to Google images for visual interpretation. A total of 120 sample points evenly distributed in the study area were selected (20 verification points for each forest land resource type). In algorithm comparison work, the overall accuracy, user accuracy, and producer accuracy are often used to determine which classifier achieves better accuracy [21,46]. In this study, we generated the overall accuracy (OA), user accuracy (UA), and producer accuracy (PA) using the validation samples. Through these accuracy measures, the performance of five algorithms for forest land resource information extraction on three feature sets was evaluated.

## 3. Results and Analysis

To assess and compare the performance of the algorithms and the different datasets, we used threes datasets (the Mul, Mul-vegetation and Mul-GLCM), and five classifiers (SVM, KNN, RF, DT and MLP algorithms). Consequently, each classifier had three classification results, totaling fifteen overall classification results.

### 3.1. Forest Land Resource Information Acquisition Results Based on Four Algorithms

Tables 4–6 show the accuracy of the forest land resource information for the three feature groups of Mul, Mul-vegetation and Mul-GLCM by SVM, KNN, RF, DT and MLP algorithms. Further analysis of Tables 4–6 found that among the three feature combinations,

the SVM algorithm achieved the highest OA (95.8%) and the DT algorithms had the lowest OA, followed by the KNN, RF and MLP algorithms; overall, the average accuracy of the SVM algorithm was higher than the other three algorithms (SVM 88.3%, KNN 87.5, RF 85.3%, MLP 85.0% and DT 77.5%). The analyses in Tables 4–6 also show that including the vegetation index on the basis of multispectral features reduces the accuracy of the SVM, KNN, RF and DT algorithms. In particular, the accuracy of the DT algorithm was reduced to 51.7%. Only the MLP algorithm improved by 2.5 percent; conversely, adding texture features can improve the accuracy of the five algorithms.

**Table 4.** Forest land resource information accuracy of five classifiers (SVM, KNN, RF, DT and MLP) based on Mul feature.

| Class | SVM | | KNN | | RF | | DT | | MLP | |
|---|---|---|---|---|---|---|---|---|---|---|
| | PA | UA | PA | UA | PA | UA | PA | UA | PA | UA |
| Broad-leaved forests | 0.750 | 0.938 | 0.600 | 0.800 | 0.800 | 0.842 | 0.750 | 0.790 | 0.500 | 0.769 |
| Shrubland | 1.000 | 0.909 | 1.000 | 0.952 | 0.950 | 0.950 | 1.000 | 0.909 | 0.950 | 0.731 |
| Barren land | 0.950 | 0.826 | 0.850 | 0.708 | 0.850 | 0.850 | 0.800 | 0.800 | 0.700 | 0.778 |
| Impervious surface | 0.950 | 1.000 | 0.900 | 1.000 | 0.900 | 1.000 | 0.900 | 1.000 | 0.900 | 0.818 |
| Grasslands | 1.000 | 0.952 | 1.000 | 0.909 | 1.000 | 0.909 | 1.000 | 0.909 | 0.800 | 1.000 |
| Coniferous forests | 0.950 | 1.000 | 1.000 | 1.000 | 1.000 | 0.952 | 0.950 | 1.000 | 1.000 | 0.800 |
| Overall Accuracy | 0.933 | | 0.892 | | 0.917 | | 0.900 | | 0.808 | |

**Table 5.** Forest land resource information accuracy of five classifiers (SVM, KNN, RF, DT and MLP) based on Mul-vegetation.

| Class | SVM | | KNN | | RF | | DT | | MLP | |
|---|---|---|---|---|---|---|---|---|---|---|
| | PA | UA | PA | UA | PA | UA | PA | UA | PA | UA |
| Broad-leaved forests | 0.000 | 0.000 | 0.400 | 0.889 | 0.550 | 0.917 | 0.300 | 0.600 | 0.400 | 0.889 |
| Shrubland | 0.700 | 1.000 | 0.800 | 0.889 | 0.050 | 0.333 | 0.000 | 0.000 | 0.800 | 0.889 |
| Barren land | 0.950 | 0.576 | 0.900 | 0.692 | 0.900 | 0.947 | 0.800 | 0.471 | 0.900 | 0.692 |
| Impervious surface | 0.900 | 0.692 | 0.900 | 0.900 | 0.900 | 0.720 | 0.050 | 0.333 | 0.900 | 0.900 |
| Grasslands | 1.000 | 0.909 | 1.000 | 0.909 | 0.950 | 0.905 | 0.950 | 0.864 | 1.000 | 0.909 |
| Coniferous forests | 1.000 | 0.800 | 1.000 | 0.800 | 1.000 | 0.500 | 1.000 | 0.392 | 1.000 | 0.800 |
| Overall Accuracy | 0.758 | | 0.833 | | 0.725 | | 0.517 | | 0.833 | |

**Table 6.** Forest land resource information accuracy of five classifiers (SVM, KNN, RF, DT and MLP) based on the Mul-GLCM.

| Class | SVM | | KNN | | RF | | DT | | MLP | |
|---|---|---|---|---|---|---|---|---|---|---|
| | PA | UA | PA | UA | PA | UA | PA | UA | PA | UA |
| Broad-leaved forests | 0.950 | 0.905 | 0.600 | 0.857 | 0.800 | 0.800 | 0.800 | 0.842 | 0.750 | 1.000 |
| Shrubland | 1.000 | 0.952 | 1.000 | 0.952 | 0.950 | 0.826 | 0.900 | 0.900 | 1.000 | 0.870 |
| Barren land | 0.900 | 1.000 | 0.900 | 0.720 | 0.800 | 1.000 | 0.850 | 0.895 | 0.950 | 0.864 |
| Impervious surface | 0.900 | 1.000 | 0.900 | 1.000 | 0.950 | 1.000 | 0.900 | 1.000 | 0.850 | 0.895 |
| Grasslands | 1.000 | 0.909 | 1.000 | 0.909 | 1.000 | 0.952 | 1.000 | 0.909 | 0.950 | 0.864 |
| Coniferous forests | 1.000 | 1.000 | 1.000 | 1.000 | 1.000 | 0.952 | 1.000 | 0.909 | 0.950 | 1.000 |
| Overall Accuracy | 0.958 | | 0.900 | | 0.917 | | 0.908 | | 0.908 | |

Table 4 shows that the RF algorithm is better than the DT algorithm in obtaining information on forest land resources of broad-leaved forests and barren land. The PA values of broad-leaved forests by the RF and DT algorithms were 80% and 75%, and those for barren land were 85% and 80%. The accuracy of the MLP algorithm was poor, at only 50% and 70%. Compared with the RF algorithm, the PA of the SVM algorithm was better for shrubland, barren land and impervious surfaces. As shown in Table 5, after the multispectral features of the Sentinel-2 image were added to the vegetation index, the accuracy rates of four out of the five algorithms, SVM, KNN, RF, and DT, decreased, with

the MLP algorithm being the exception. The shrubland PA value of the RF algorithm was 65% and 75% lower than that of the SVM and KNN algorithms, respectively. In addition, the SVM, KNN and RF algorithms mainly produced improvements in PA values for impervious surfaces, grasslands and coniferous forests. As shown in Table 6, after adding texture features on the basis of multi-spectral features, and compared with adding vegetation index, the five algorithms have improved the classification effect of six types of ground object. At the same time, the accuracy of the SVM algorithm is better than for the other four algorithms in some categories.

In summary, the SVM algorithm obtained high extraction accuracies for land-use types, and KNN and MLP have better robustness on three feature sets. Adding texture features on the basis of multispectral features can effectively improve the classification accuracy of ground object types, but the accuracy is reduced by adding the vegetation index.

The spatial distribution of forest land resource information based on five classifiers of Mul, Mul-vegetation and Mul-GLCM is shown in Figures 3–5.

- **The spatial distribution of forest land resource information based on five classifiers based on Mul:**

Based on the Mul features, SVM better identified forest land resource types that were difficult for other algorithms to distinguish, including barren land and impervious surface. However, the forest land resource-type extraction effect of KNN, RF, DT and MLP algorithms was unsatisfactory. Further analysis of Figure 3 shows that the SVM algorithm can correctly distinguish between barren land and impervious surface, while the KNN, RF, DT and MLP algorithms confused them; the RF algorithm accurately identified broad-leaved forests and shrubland, while KNN, MLP and DT misclassified broad-leaved forests as shrubland; barren lands were misclassified as broad-leaved forests for KNN, RF, DT and MLP while the SVM algorithm accurately distinguished between them; SVM, KNN, RF and DT algorithms misclassified some grasslands as barren lands, and MLP misclassified some grasslands as impervious surface.

- **The spatial distribution of forest land resource information based on five classifiers based on Mul-vegetation:**

Based on the Mul-vegetation features, the KNN and MLP algorithms better recognized shrubland, barren land and impervious surface. On the contrary, the recognition effect of SVM, RF and DT algorithms was poor. As shown in Figure 4, RF and DT algorithms confused broad-leaved forests, shrubland and barren land, whereas the KNN and MLP algorithms correctly distinguished shrubland and barren land. The DT algorithm misidentified the impervious surface, while the other four algorithms had better identification results of forest land resource information. SVM, KNN and MLP identified shrubland and coniferous forests, but RF and DT algorithms misclassified shrubland as coniferous forests. All the classifiers misclassified some broad-leaved forests as shrubland, barren land, impermeable surface and coniferous forests.

- **The spatial distribution of forest land resource information based on five classifiers based on Mul-GLCM:**

Based on Mul-GLCM, five algorithms can better identify each category of forest land resource type, especially grassland and coniferous forests. These identification results are the best. As shown in Figure 5, the KNN, RF, DT and MLP algorithms incorrectly classified most broad-leaved forests as shrubland and barren land, whereas the SVM algorithms identified these types better; the SVM algorithms accurately identified the broad-leaved forests, while the KNN, RF, DT and MLP algorithms misclassified them; all classifiers misidentified some impervious surfaces as grassland.

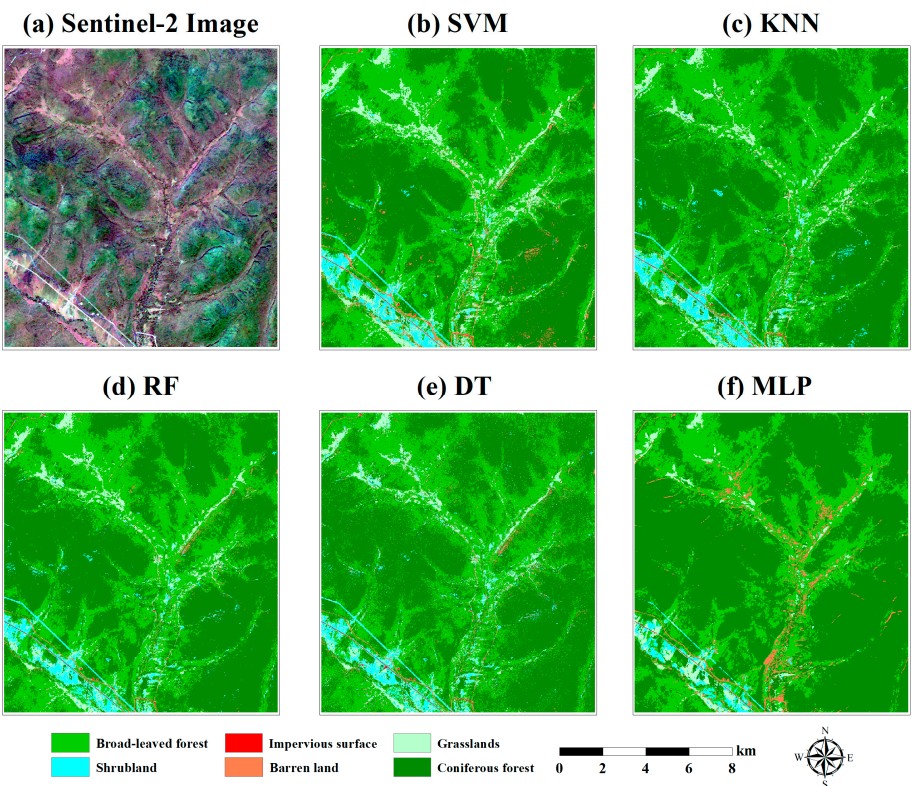

**Figure 3.** Sentinel-2 remote sensing image (**a**) and forest land resource information maps of the five-classifiers based on Mul features: (**b**) SVM classifier; (**c**) KNN classifier; (**d**) RF classifier; (**e**) DT classifier; (**f**) and MLP classifier.

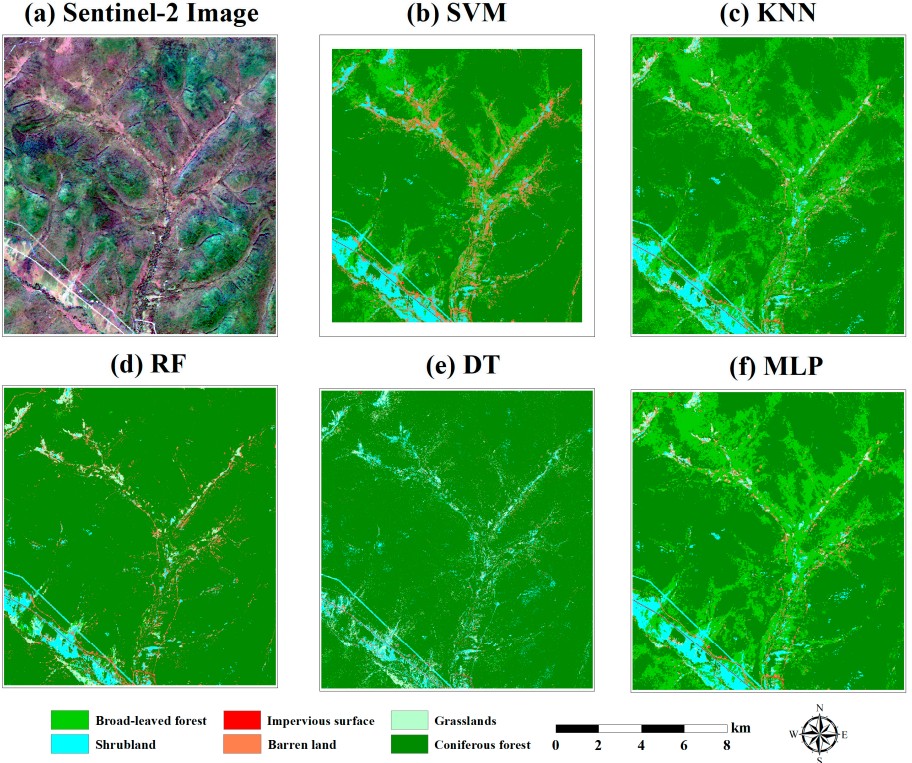

**Figure 4.** Sentinel-2 remote sensing image (**a**) and forest land resource information maps of the five-classifiers based on Mul-vegetation features: (**b**) SVM classifier; (**c**) KNN classifier; (**d**) RF classifier; (**e**) DT classifier; (**f**) and MLP classifier.

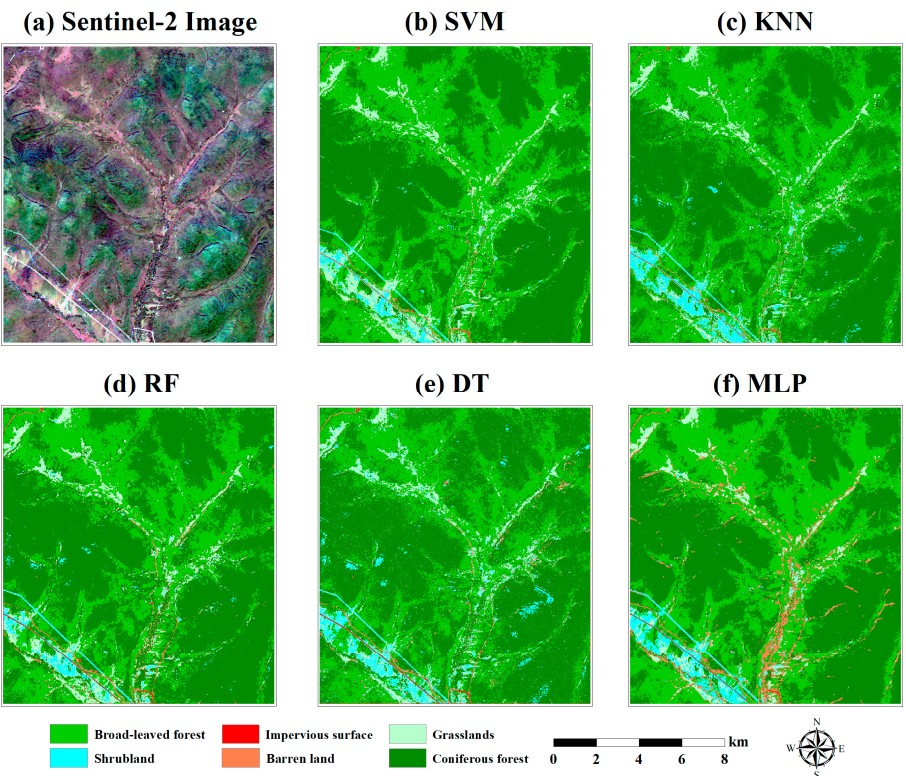

**Figure 5.** Sentinel-2 remote sensing image (**a**) and forest land resource information maps of the five-classifiers based on Mul-GLCM features: (**b**) SVM classifier; (**c**) KNN classifier; (**d**) RF classifier; (**e**) DT classifier; (**f**) and MLP classifier.

### *3.2. Forest Land Resource Information Acquisition Confusion Matrix Results Analysis*

As shown in Figure 6, a confusion matrix was constructed based on the forest land resource information extraction results of SVM, KNN, RF, DT and MLP algorithms on Mul, Mul-vegetation and Mul-GLCM. The current study found that when using the combination of three features to extract information on forest resources, each algorithm had a relatively high classification accuracy for coniferous forests, and the recognition accuracy was above 95%, while the accuracy of other forest land resource categories varied greatly.

Further analysis of Figure 6 found that (1) based on the Mul features, the forest land resource information extraction accuracies of the SVM algorithm for Barren land were significantly better than for the KNN, RF, DT and MLP algorithms, whereas the DT and MLP algorithms misclassified barren land as broad-leaved forests at a higher rate. The KNN mistakenly classified 35% of the broad-leaved forests as barren land, while the SVM, DT and MLP algorithms only predicted 20% incorrectly. In conclusion, the SVM algorithm has obvious advantages in forest land resource information acquisition. (2) For the Mul-vegetation features, while the accuracy of SVM, KNN, RF and DT algorithms all have different degrees of decline, MLP improved. Five algorithms have a higher classification accuracy for grasslands and coniferous forests, with an accuracy of 95% and 100%. Compared with those of the RF and DT algorithms (95%), the SVM, KNN and MLP algorithms yielded higher extraction accuracies of Grasslands (100%). A total of 95% of shrubland was classified into coniferous forests by the RF and DT algorithms, while the misclassification rates of SVM, KNN and MLP algorithms were only 25% and 20%. Therefore, the KNN and MLP algorithms performed the best at collecting forest land resource information compared with other algorithms. (3) For Mul-GLCM features, the SVM and KNN algorithms had very high accuracy for shrubland, grasslands, and coniferous forests (100%), which is the highest among all grassland classification results. Compared with the forest land resource information acquisition accuracy of KNN, RF and DT (65%, 80%), the accuracy of SVM is 15–35% higher. From the data in Figure 6, it

is apparent that the RF and DT algorithms misclassified 20% of the broad-leaved forests as shrubland and barren land, and the misclassification rate of KNN is 5% and 35%, respectively. MLP also reaches 25%, whereas the misclassification rates of SVM were relatively low (5%).

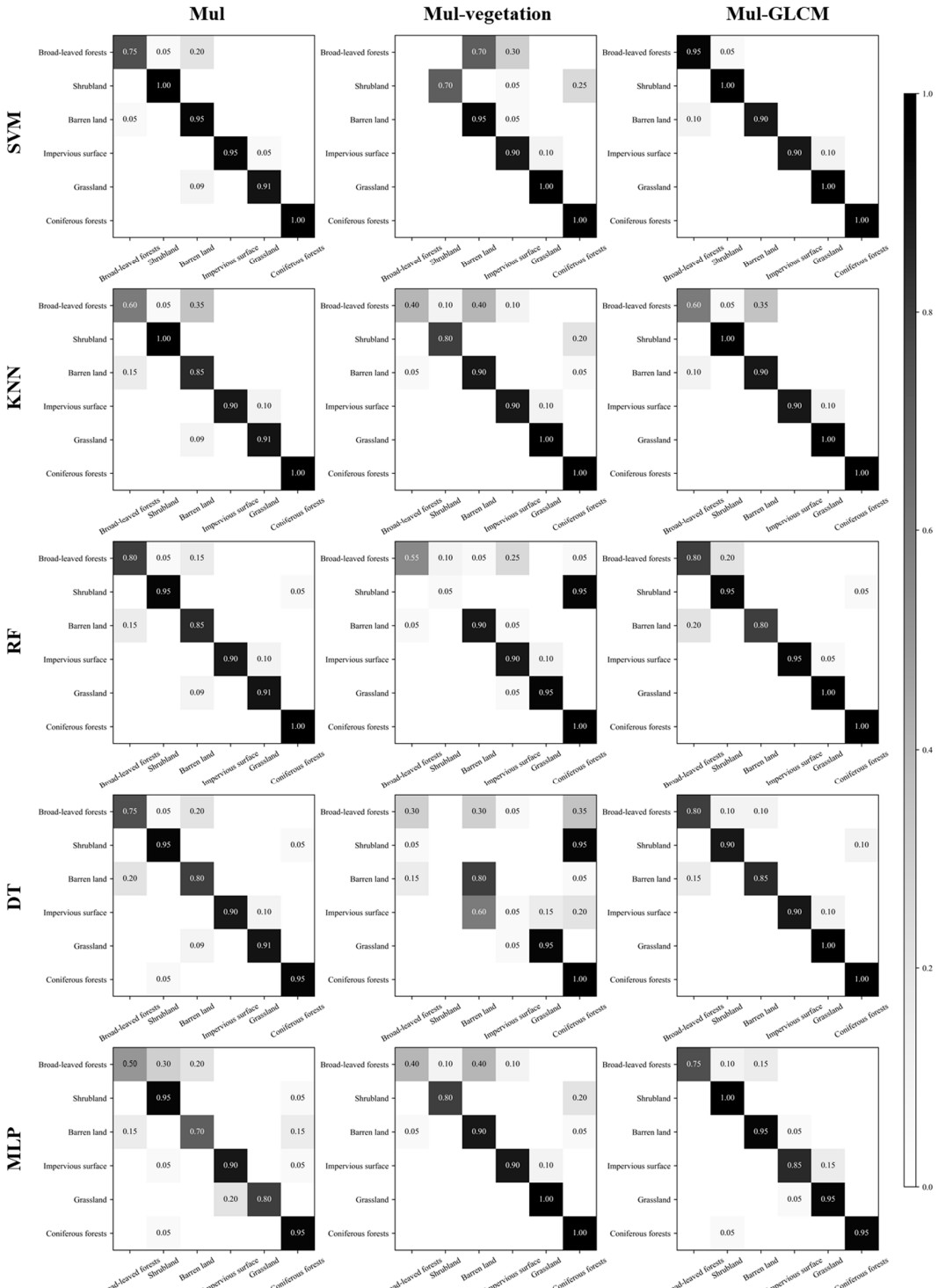

**Figure 6.** There are four classification algorithms of SVM, KNN, RF, DT and MLP in the horizontal direction, three features of Mul, Mul-vegetation and Mul-GLCM in the vertical direction, and a total of 15 confusion matrix results.

## 4. Discussion

The classification of Sentinel-2 imagery using five algorithms was implemented, evaluated, and compared. Three different datasets, including Mul features, Mul-vegetation features and Mul-GLCM features, were used.

Figure 7 shows the difference between the OA of datasets from the 15 results of three different training sample sizes for five classifiers. Two different trends are clear: Compared with the Mul features, (1) the accuracy of the four of the five algorithms, SVM, KNN, DT and MLP, improved in the Mul-GLCM features, not including the RF algorithm; in the Mul-vegetation feature set, four algorithms showed different degrees of degradation, although the MLP algorithm did not. (2) KNN and MLP showed good stability on the three datasets. However, the performance of SVM, KNN, RF, DT and MLP on three datasets was significantly different.

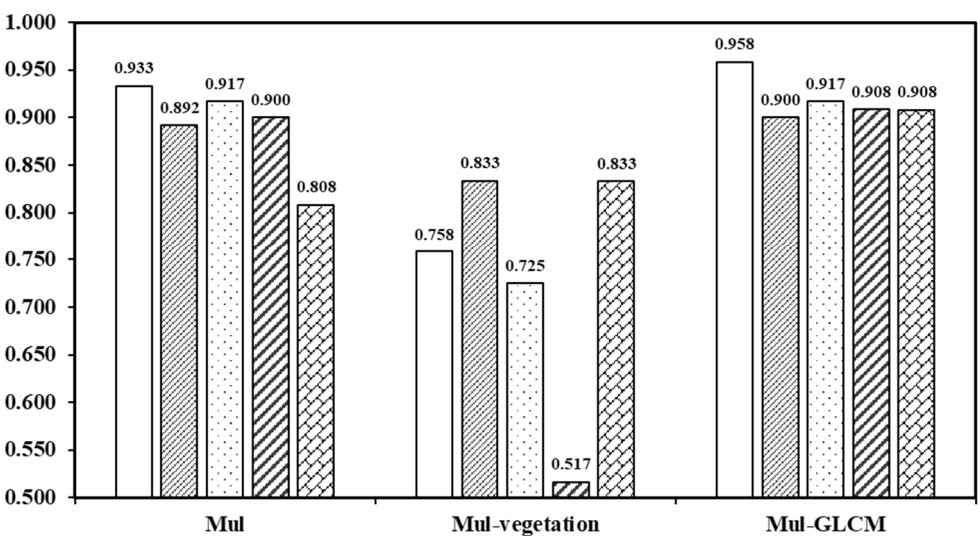

**Figure 7.** Forest land resource information Overall accuracies (OA) of three features based on four algorithms.

As shown in Figure 7, the SVM algorithm achieved the highest OA (95.8%). The average accuracy of the SVM algorithm was much higher than the other algorithms (SVM 88.3%, KNN 87.5, RF 85.3%, MLP 85.0%, DT 77.5%). The analysis in Tables 4–6 also shows that adding the vegetation index onto the basis of multispectral features causes a decline in the accuracy of each algorithm, especially the accuracy of the DT algorithm, which reduced to 51.7%. Previous studies have shown that NDVI can significantly reflect vegetation cover and plant physiological status [47]; GRVI can reflect plant growth and health [48,49]; RVI can enhance the difference between vegetation and soil background radiation values, and can better distinguish vegetated areas from non-vegetated areas in areas with high vegetation cover [50]; DVI is highly correlated with vegetation soil background variation values and is often used to distinguish vegetation from water bodies [47]; NDREI is sensitive to small changes in vegetation parameters [51]; and LSWI can sense changes in plant and soil moisture content and is more effective when used to distinguish between different vegetation types [52,53]. This is quite different from previous research results. Therefore, we further investigated the distribution of the six taxonomic categories over the respective vegetation. Figure 8 shows that the separation degree of the six classification categories is relatively large for DVI, and the distribution of other vegetation indices is relatively concentrated, meaning the difference is not obvious. The majority of the vegetation indices overlapped considerably in different forest land resources, and the separation was not obvious enough to be used as a feature to distinguish between different land-use types. Careful consideration should be given when considering vegetation indices.

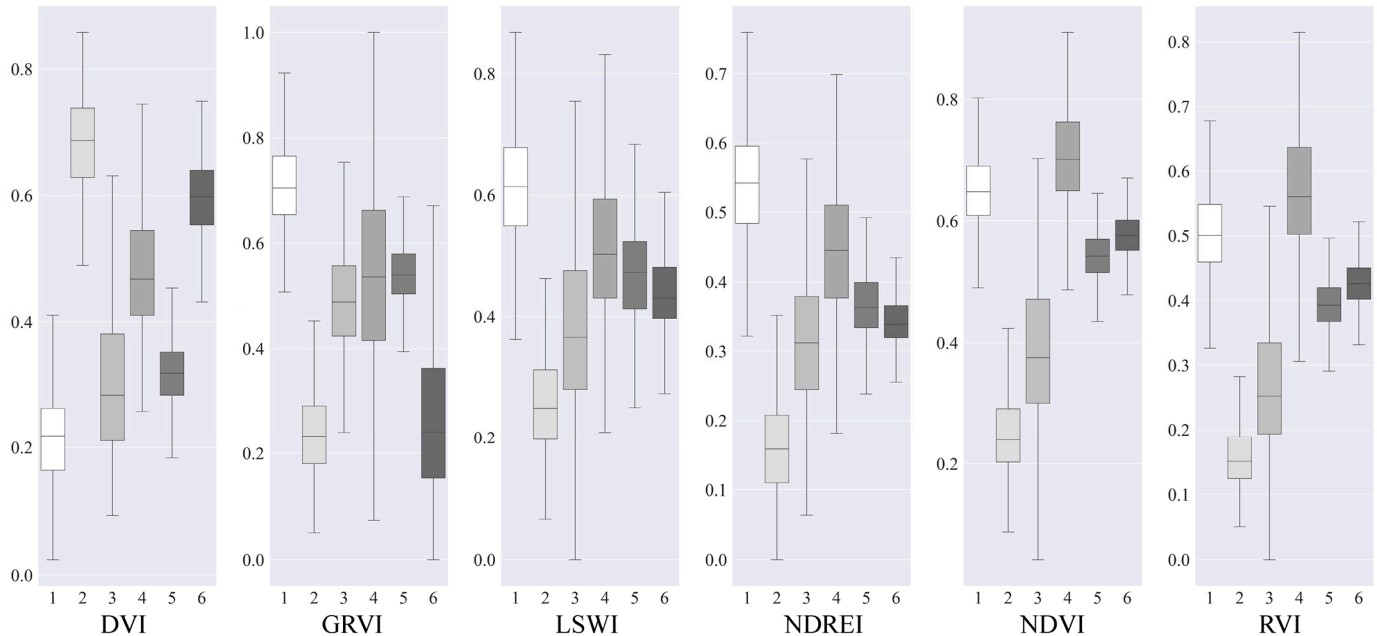

**Figure 8.** Boxplot of each category on the vegetation index. The centerline in each box in the boxplot is the median, and the edges of the box represent the upper and lower quartiles. Abscissa label: 1. Coniferous forests; 2. Impervious surface; 3. Barren land; 4. Shrubland; 5. Broad-leaved forests; 6. Grasslands.

In this study, the vegetation index may lead to lower forest land resource information acquisition accuracy in multispectral bands. However, the combination of multispectral features and GLCM texture features improved the accuracy more than just by using multispectral features. Comparing our results with the results of other studies confirms that texture feature extraction plays a very important role in improving the classification accuracy of remote sensing images. This supports other studies in this area by linking multispectral features with GLCM texture features. This finding is consistent with that of Zhang et al. [54], who verified the importance of GLCM texture feature extraction as a classification accuracy improvement. These texture features provide information about different objects with the same spectra, while spectral bands provide the data for the same objects in different spectra. These results reflect those of Zheng et al. [55], who also found that extracted and added GLCM textures can improve the discrimination between categories of shrubland, agricultural land, and barren areas. Further analysis shows that the forest land resource information acquisition accuracies of the SVM and KNN algorithms for shrubland, grasslands, and coniferous forests were very high (100%) on the Mul-GLCM features, and these accuracies were the highest among all the grassland classification results. Compared with Mul features and Mul-vegetation features, on Mul-GLCM features, the average classification accuracy of the five algorithms for broad-leaved forests was the highest (78%, 68% and 33%). Therefore, it provides a reference for feature sets mapping forest land resources information from Sentinel-2 imagery.

It is apparent from this study that the performance of SVM, KNN, RF, DT and MLP on Mul features, Mul-vegetation features and Mul-GLCM features was significantly different. However, the single most striking observation to emerge from the data comparison was that KNN and MLP showed good stability on the three datasets. There are several possible explanations for this result. In this study, MLP would have a better recognition rate and a faster classification speed. However, its training is not as fast as with SVM classification, especially for a huge training set. If the time required for classification is great, the MLP method is a good choice. The MLP model neural network in this study is composed of multiple hidden layers, and the different layers are fully connected. The fully connected layer maps the original data to the hidden layer feature space (feature extraction + selection

process); it maps the learned feature representation to the label space of the sample. Then, the features are highly refined (integrated together) to improve the model performance. Results of this study show that its network structure is reasonable and its model is robust. Therefore, recent MLP-based work shows that even a structure that is simple in terms of design philosophy and design skills can achieve comparable performance to CNN and Transformer [56]. One of the advantages of the KNN classifier is that it has few parameters, all of which are intuitive. At the same time, the KNN classifier can work with very little training data. KNN was the fastest of all classifiers during training. KNN directly compares each unknown sample with the original training data [57,58], which is a more extreme form of the instance-based method, and the comparison between samples is kept as part of the model during all training processes. Inside the model, predictive decisions are made by using the rivalry between data instances. The objective similarity measure between samples is to compare the similarity between the given sample data and unknown data, which is helpful for the prediction of the model. The KNN model does not build a model from the given training samples until it needs to make a prediction. In this way, a timely response can be made based on the corresponding actual samples.

The findings based on the above results give us more thinking, and there is still a lot of work to be done in the future. On the one hand, neural network models have great advantages in solving complex problems, and should be vigorously developed in the field of forestry to make forestry develop towards intelligence; on the other hand, machine learning and deep learning have their own advantages. How to design a reasonable model, and find an economical, efficient and accurate method of obtaining information on forest land resources requires further research.

## 5. Conclusions

Using Sentinel-2 satellite multispectral image data, the spectral reflectance, vegetation index characteristics and image texture characteristics of different forest land resources in the study area were calculated and compared, and then, based on three groups of features (Mul features, Mul-vegetation features and Mul-GLCM features), five classification algorithms, SVM, KNN, RF, DT and MLP, were constructed to identify and classify forest land site types. The research indicates: (1) Among the three feature combinations, the SVM algorithm achieved the highest OA (95.8%). The average accuracy of the SVM algorithm was much higher than other algorithms (SVM 88.3%, KNN 87.5%, RF 85.3%, MLP 85.0%, DT 77.5%). (2) The classification accuracies of each algorithm for coniferous forests were relatively high, and the recognition accuracy was above 95%, whereas the classification accuracies of the other categories varied greatly. (3) The results show that adding texture features can improve the accuracy of the algorithms; on the contrary, adding vegetation index based on multispectral features reducing the accuracy of each algorithm. In particular, the accuracy of the DT algorithm was reduced to 51.7%. This study provides a new reference for the qualitative methods of forest land resource distribution. It has also produced more efficient and accurate acquisition of forest land resource information, scientific management and effective use of forest land resources.

**Author Contributions:** Conceptualization, C.Z.; Data curation, C.Z.; Formal analysis, C.Z.; Funding acquisition, N.T.; Investigation, C.Z.; Methodology, C.Z.; Project administration, N.T.; Resources, C.Z.; Supervision, Y.L. and N.T.; Validation, C.Z.; Visualization, C.Z.; Writing—original draft, C.Z.; Writing—review and editing, C.Z. and Y.L. All authors have read and agreed to the published version of the manuscript.

**Funding:** This research was funded by the Science and Technology Planning Project of Inner Mongolia Autonomous Region, grant number No. 2020GG0067.

**Data Availability Statement:** The data presented in this study are available on request from the first author.

**Acknowledgments:** This research was funded by the Science and Technology Planning Project of Inner Mongolia Autonomous Region, grant number No. 2020GG0067. We thank Fan Yang, from the Institute of Ecology and Forestry, Sichuan Agricultural University for editing the English text of a draft of this manuscript.

**Conflicts of Interest:** The authors declare that they have no competing interests.

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
