# Peer review of "Forest Land Resource Information Acquisition with Sentinel-2 Image Utilizing Support Vector Machine, K-Nearest Neighbor, Random Forest, Decision Trees and Multi-Layer Perceptron"

_forests, doi:10.3390/f14020254_

Round 1
Reviewer 1 Report
This is a competent manuscript on methods to improve the accuracy of mapping different forest landscapes using remote sensing. As the authors discuss, studies of improved delineations can be accomplished by comparing data sets or processing strategies and they do both. This parallel effort required considerable processing but also created a relatively complex manuscript. The study adds to the literature but not significantly as it is one more of relatively similar studies.
The manuscript has a clear statement of intent and a logical flow. The tables and figures are professional and informative. There is a very extensive set of appropriate references and with a few possible exceptions, in complete and consistent format.
The manuscript is generally quite well written but as most manuscripts, would benefit from another editorial review. Some suggestions for consideration by the authors follow:
1. There is inconsistent use of serial commas.
2. Acronyms need to be defined and then employed typically in both the abstract and the main text. There are many violations of this.
3. Line 13 and elsewhere, data are plural.
4. Line 18, awkward sentence.
5. Line 22, four are stated but only three listed?
6. Line 39, long, complex, confusing sentence.
7. Line 44, lot is poor word choice.
8. Line 40, resolution requires a modifier such as spatial, temporal etc.
9. Line 56, et al requires a period.
10. The community generally employs the term Land Use Land Cover (LULC) which the authors might consider as they are not consistent.
11. Line 70, drawing is a poor word choice.
12. Line 74, research.
13. Line 76, state country.
14. Is Figure 1 cited in text?
15. The remote sensing literature typically employs the terms calibration and validation for training and truth. The authors might follow that or at least be consistent. There needs to be a clear statement that the calibration and validation points are independent. The authors might also discuss the very low number of validation points (100).
16. This reviewer ceased making editorial suggestions after about line 100.
17. Reference 7 appears incomplete. Why is 14 in upper case?
As stated this is a viable contribution to the literature. It requires some clarifications and editorial review. The selection of this journal, Forests, will bring the information to a different community than a remote sensing journal which is a contribution.
Author Response
Dear reviewer,
We sincerely acknowledged for your kind critical comments and suggestions in improving the quality of the manuscript. We have carefully revised the manuscript.
The following is response to your comments and suggestions.
Sincerely,
Chen Zhang,
Inner Mongolia Agriculture University, College of Forestry
Email: zhangchenstudy@163.com

Reviewer 2 Report
Dear Authors,
Thank you for considering Forests for your work. I would like to admit that the study is well performed - the study design is well set up too. However, with the advent with other machine learning and deep learning algorithms, the community is striving to improve accuracy performance by employing new methods. Despite the study merits, I would like to suggest you to either justify you do not consider deep learning in your research or to extend it with a deep learning approach. That will bring value and novelty of your work and make it contemporary. Thank you in advance for your consideration.
Kind regards,
Reviewer
Author Response

(The authors gave the same response as above.)
